# Calibration under Sparse Data: Robust Canonical Surface Estimation from Transaction Bars

**Ekaterina Bagantsova**
Faculty of Computer Science
National Research University Higher School of Economics
11 Pokrovsky Bulvar, Moscow, 109028, Russia
`erbagantsova@edu.hse.ru`

## Abstract

We study intraday canonicalization and auditing of implied-volatility (IV) surfaces for S&P 500 index (SPX) options when the available observations are sparse, trade-based open–high–low–close–volume (OHLCV) bars. Such panels exhibit strongly non-uniform cross-sectional support—dense near at-the-money (ATM) and short maturities, and thin in the wings and long tenors—so naive fixed-grid completion can create visually smooth surfaces with unstable calibration and misleading feasibility diagnostics. We propose a reliability-aware canonicalization pipeline that (i) constructs forward and discount proxies from liquid futures strips, (ii) quantifies local information content via an effective sample size (ESS) diagnostic induced by kernel receptive fields, and (iii) calibrates a per-timestamp Surface SVI (SSVI) total-variance surface directly in price space under robust losses. Our main methodological contribution is a reliability-weighted vega-normalized robust objective that downweights weakly supported marks while retaining the interpretability and tractability of price-space calibration. We compare three calibration objectives—robust price residuals, vega-normalized residuals, and reliability-weighted vega residuals—using paired timestamp-level inference with Holm-adjusted randomization tests, and we evaluate economic plausibility with a unified static-arbitrage audit that reports both violation rates and hinge-type severity measures. Empirically, price-residual calibration minimizes price root-mean-squared error (RMSE), whereas reliability-weighted vega residuals yield the most consistent reductions in out-of-sample (OOS) arbitrage severity and in average any-rule violation rates. These results support reliability-weighted robust objectives as default canonicalizers for economically plausible intraday surfaces built from sparse transaction bars and for downstream learning tasks.

## 1 Introduction

Intraday option markets encode a high-dimensional, rapidly evolving surface of state-contingent prices across strike and maturity. In many practical data feeds and brokerage-grade interfaces, however, intraday observations are delivered as trade-based bars rather than synchronized quote books, inducing strong missingness, heterogeneous staleness, and uneven coverage over the strike–maturity plane. Recent empirical evidence highlights both the scale and structural complexity of modern implied-volatility surfaces (IVSs) in large option universes (Ulrich et al., 2023), while recent forecasting studies treat the IVS as a dynamic object and document substantial nonlinear dependence across maturities and moneyness (Medvedev & Wang, 2021; Chen et al., 2025). These advances largely assume that the surface is observed sufficiently densely (often at daily frequency), whereas intraday transaction-bar sampling produces a markedly different regime in which local support can be sparse precisely where extrapolation risk is highest.

A second, tightly coupled obstacle is economic feasibility. Predictive accuracy alone does not guarantee economically plausible surfaces, motivating approaches that separate prediction from arbitrage-free reconstruction (Zhang et al., 2022) and methods that learn or simulate surfaces while

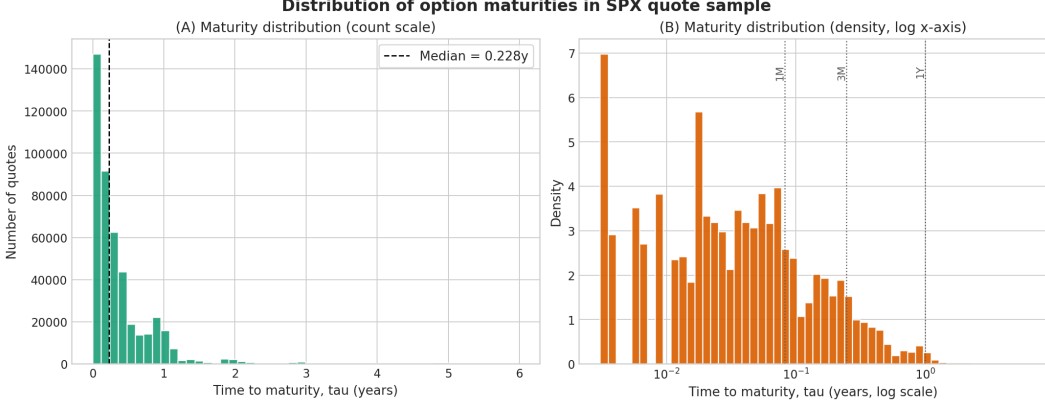

Figure 1: Distribution of option maturities in the SPX sample. Panel (A) reports histogram counts with the median marker. Panel (B) shows the same distribution on a log-maturity axis to highlight short-end concentration, with vertical guides at 1M, 3M, and 1Y.

controlling static arbitrage (Cont & Vuletić, 2023; Ning et al., 2023; Vuletić & Cont, 2024). Parallel work on joint underlying–IVS models (Francois et al., 2025), structured parametrizations (Zaugg et al., 2025), and hybrid model-plus-learning corrections (Duan et al., 2025) further underscores that data sparsity and irregular coverage materially affect reconstruction quality (Shao et al., 2025).

This paper targets a pragmatic setting that is not well served by quote-centric assumptions: intra-day trade-bar panels with heterogeneous activity and strongly non-uniform local support. With $t$ an intraday timestamp, $T$ a calendar maturity date, and $\tau = \tau(t,T)$ time to maturity, let $k = \log\big(K/F(t,T)\big)$ denote log-moneyness. Our goal is to produce, at each timestamp, a *canonical surface*: a complete, deterministic total-variance grid $w(k,\tau)$ on a fixed $(k,\tau)$ mesh, paired with a per-cell reliability indicator, that serves as a reproducible learning target while retaining a direct, auditable link to traded information. The key methodological challenge is that canonicalization necessarily mixes interpolation with extrapolation; therefore, a single "completed" surface is insufficient without an explicit, quantitative account of which regions are supported by data. This reliability perspective is complementary to recent work on IVS dynamics and structure across asset classes (Alfeus et al., 2024) and to modern econometric treatments of nonlinear surface dependence (Chen et al., 2025).

Our approach is reliability-aware by design. First, we construct forward and discount proxies from liquid futures strips to support a consistent forward–discount pricing map. Second, we quantify local support through an effective-sample-size (ESS) diagnostic induced by kernel receptive fields and complement it with activity proxies (volume and trade count), yielding observation-level reliability weights. Third, we calibrate a Surface SVI (SSVI) total-variance surface per timestamp directly in price space under robust losses, comparing three objectives that differ only in residual normalization and reliability weighting. Fourth, we apply a unified static-arbitrage audit to raw marks, kernel-completed synthetic quotes, and SSVI synthetic quotes, reporting both violation rates and hinge-type severities.

We do not propose a new parametric smile family; our contribution is a reliability-aware canonicalization protocol tailored to sparse transaction bars that explicitly propagates local data support into calibration and evaluation. Concretely: (i) a reliability layer for intraday panels based on ESS induced by kernel receptive fields and activity proxies; (ii) a robust price-space SSVI calibration strategy with a reliability-weighted vega-normalized objective; (iii) a unified feasibility audit that reports both incidence and magnitude of static-arbitrage violations; and (iv) a paired timestamp-level comparison protocol with multiple-testing control. In empirical results, constant-scale price residuals minimize price RMSE, whereas reliability-weighted vega residuals yield the most consistent out-of-sample reductions in static-arbitrage severity. The appropriate default depends on the downstream objective: when surface plausibility and downstream learning stability are prioritised, `wvega` is the recommended choice; when price-fit accuracy is paramount, `simple` remains preferable (Section 7).

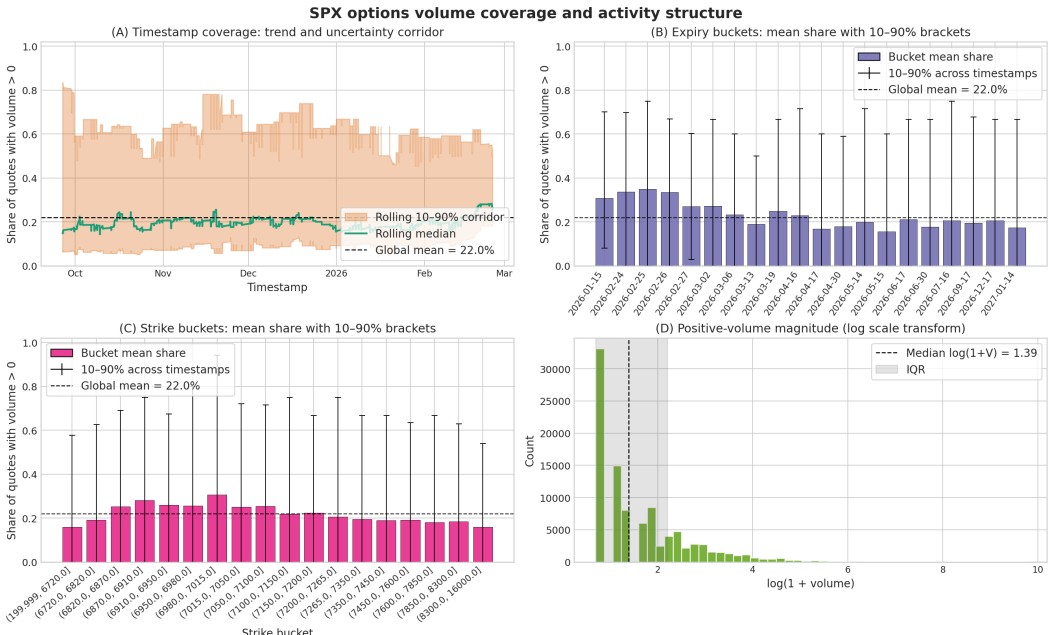

Figure 2: SPX options volume coverage and activity structure. Panel (A) shows the timestamp-level share of positive-volume bars with a rolling 10–90% corridor and rolling median. Panels (B) and (C) report mean positive-volume share by expiry and strike buckets with 10–90% across-timestamp brackets. Panel (D) shows the distribution of $\log(1 + \text{volume})$ for positive-volume bars.

## 2 DATA AND MARKET MICROSTRUCTURE

### 2.1 DATA UNIVERSE

We consider three synchronized intraday panels sampled at a fixed 30-minute frequency: (i) SPX option contracts (calls and puts across strikes and expiries), (ii) E-mini S&P 500 index futures (ES) as a proxy for the equity forward and market state, and (iii) three-month SOFR futures (SR3), where SOFR denotes the Secured Overnight Financing Rate, used as a tradable proxy for short-term funding and discounting inputs.

### 2.2 TRADE-BAR CLEANING AND MISSINGNESS

Trade bars can be internally inconsistent due to feed glitches, partial updates, or session boundaries. Writing $(O, H, L, C)$ for open/high/low/close, we apply minimal feasibility filters:

$$O > 0, \quad C > 0, \quad H \geq L, \quad O \in [L, H], \quad C \in [L, H].$$

Option bars frequently exhibit zero traded volume and/or zero trade count, reflecting sparse trading. In surface construction, we treat such intervals as missing observations for price inference (while retaining them for coverage statistics). Figure 1 shows that maturity mass is concentrated at short tenors, motivating heteroskedastic treatment across $\tau$. Figure 2 indicates persistent but non-uniform liquidity coverage across time, expiry, and strike dimensions.

### 2.3 NEAR-EXPIRY INSTABILITY AND MATURITY FLOOR

For an option with maturity $\tau$, implied-volatility estimation is sensitive to price noise, particularly as $\tau \to 0$. This follows from the first-order relationship between price perturbations and implied-volatility perturbations, which scales inversely with Black vega, and is documented empirically in Hentschel (2003). We therefore impose a minimum maturity threshold $\tau \geq \tau_{\min}$, chosen to stabilize intraday inference and to avoid extreme microstructure amplification at the shortest maturities.

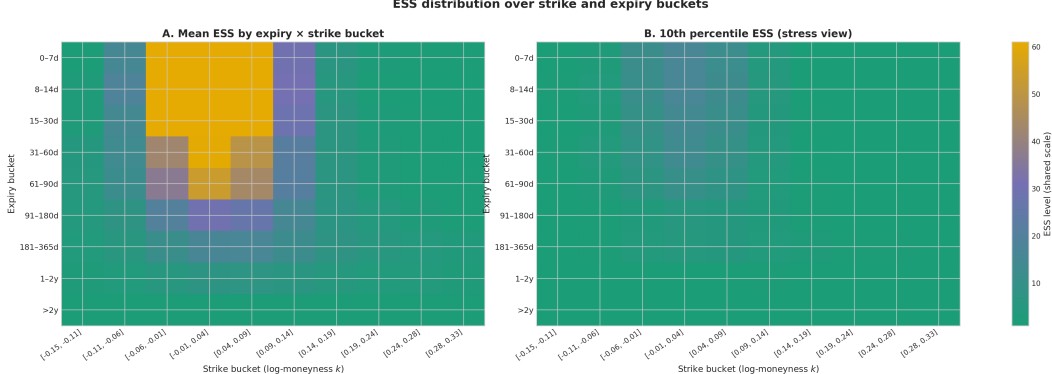

Figure 3: Effective sample size (ESS) across strike and expiry buckets. Panel (A) reports mean ESS, and Panel (B) reports 10th-percentile ESS under the same color normalization. Higher ESS indicates denser local neighborhoods used by the kernel smoother.

## 2.4 FORWARD AND DISCOUNT PROXIES

To map strikes to log-moneyness and to implement forward-discount Black inversion, we require proxies $F(t,T)$ and $D(t,T)$. We construct $(F(t,T))$ from the ES futures strip by monotone maturity ordering and log-linear interpolation within the liquid knot range, and we cap maturities beyond the last reliable futures knot. Discount factors $(D(t,T))$ are bootstrapped from SR3 contracts across International Monetary Market (IMM) accrual windows and interpolated log-linearly.

# 3 RELIABILITY WEIGHTS AND OPTIONAL MASK CHANNEL

## 3.1 WHY RELIABILITY MATTERS

Intraday option observations induce an uneven sampling design over $(k,\tau)$: liquid regions (near ATM and short maturities) are dense, whereas wings and long maturities are thin. Any completion method therefore combines interpolation with extrapolation. Treating completed surfaces as uniformly reliable can overemphasize weakly supported regions in calibration and feasibility diagnostics. We therefore attach a reliability layer to the pipeline: observation-level weights and an optional grid-level mask.

## 3.2 ESS FROM KERNEL RECEPTIVE FIELDS

We define a fixed $(k,\tau)$ grid and quantify local support using kernel receptive fields in $(k,\log\tau)$:

$$d_i^2 = \left(\frac{k_i - k_0}{b_k}\right)^2 + \left(\frac{\log\tau_i - \log\tau_0}{b_{\log\tau}}\right)^2, \qquad (1) \qquad\qquad \omega_i = e^{-\frac{1}{2}d_i^2}, \qquad (2)$$

where $(k_0, \tau_0)$ denotes the grid cell of interest, $\{(k_i, \tau_i)\}_i$ are observed contracts at the timestamp, and $(b_k, b_{\log\tau})$ are bandwidths.

The effective sample size (ESS) at a grid cell is

$$\text{ESS} = \frac{\left(\sum_i \omega_i\right)^2}{\sum_i \omega_i^2}, \qquad (3)$$

a standard degeneracy measure for weighted averages Kish (1965). Bandwidths $(b_k, b_{\log\tau})$ are selected adaptively per timestamp from the data span (see Appendix F for a sensitivity analysis). Figure 3 reports mean ESS and 10th-percentile ESS (stress view), highlighting strong non-uniformity of support across the domain. Figure 4 illustrates how the smoother adapts to local data density.

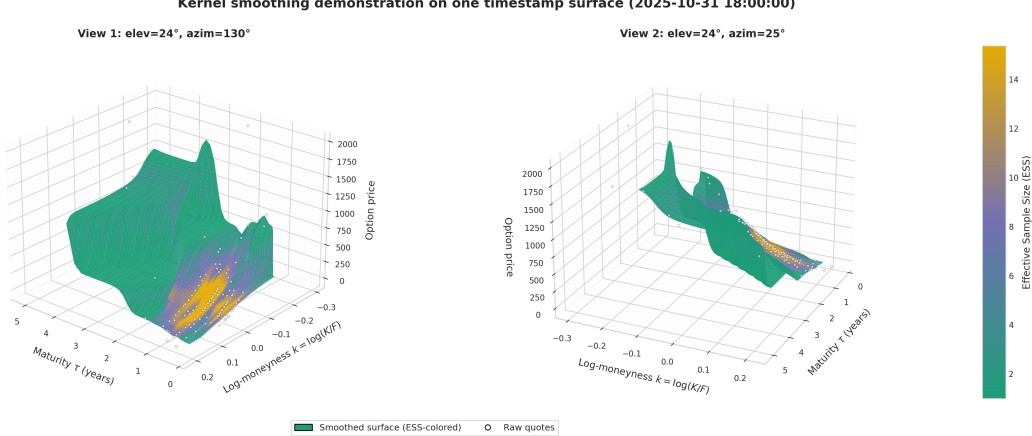

Figure 4: Kernel-smoothed option surface at a representative timestamp, shown from two viewing angles. Surface color encodes ESS; optional point-cloud overlays show raw marks. The visualization jointly illustrates smoothness and local data support.

## 3.3 COMPOSITE RELIABILITY WEIGHT

We combine ESS-based support with activity proxies. Let $V_i$ be traded volume and $N_i$ be trade count for bar $i$. When staleness is available, let $s_i^*$ be elapsed time since last trade. Define

$$w_{\text{act},i} = \log(1 + V_i) \cdot \log(1 + N_i), \qquad (4) \qquad\qquad w_{\text{stale},i} = e^{-\lambda s_i^*}, \qquad (5)$$

and

$$w_{\text{ess},i} = \begin{cases} \sqrt{\dfrac{ESS_i}{ESS_0}}, & ESS_i \leq ESS_0 \\ 1, & ESS_i > ESS_0, \end{cases} \qquad (6) \qquad w_{\text{base},i} = w_{\text{act},i} \cdot w_{\text{ess},i} \cdot w_{\text{stale},i}, \quad (7)$$

with user-chosen thresholds $\text{ESS}_0 > 0$ and $\lambda \geq 0$ (take $\lambda = 0$ if staleness is unavailable). The square-root form of $w_{\text{ess},i}$ is motivated by the classical relationship between the variance of a kernel-weighted mean and local sample size: the standard error of a weighted average scales as $1/\sqrt{\text{ESS}}$, so taking $\sqrt{\text{ESS}_i/\text{ESS}_0}$ maps local support onto a variance-proportional reliability scale and saturates at 1 once ESS reaches $\text{ESS}_0$. We stress that ESS measures local data support under the chosen kernel geometry, not ground-truth reliability; the sensitivity analysis in Appendix F confirms that conclusions are robust to bandwidth perturbations. The logarithmic transform in $w_{\text{act},i}$ compresses heavy-tailed volume and count distributions; an ablation study (Appendix H) confirms that this multiplicative log design achieves the lowest OOS vega-RMSE among four candidate formulas, indicating that the interaction between volume and bar count most effectively down-weights illiquid quotes in the vega-weighted objective. The composite base weight is bounded and normalized per timestamp:

$$\hat{w}_{\text{base},i} = \begin{cases} L_D, & w_{\text{base},i} < L_D \\ w_{\text{base},i}, & L_D \leq w_{\text{base},i} < L_U \\ L_U, & w_{\text{base},i} \geq L_U \end{cases} \qquad (8) \qquad \widetilde{w}_{\text{base},i} = \dfrac{\hat{w}_{\text{base},i}}{\frac{1}{N_t}\sum_{j=1}^{N_t} \hat{w}_{\text{base},j}}. \quad (9)$$

In this paper, weights enter calibration (Section 5) and reliability-stratified diagnostics. A binary mask $M_t(k, \tau) = \mathbf{1}\{\text{ESS}(k, \tau) \geq \text{ESS}_0\}$ can be stored for downstream learning tasks but is not required for the calibration results reported here.

## 4 CANONICAL SURFACE MODEL: SSVI PER TIMESTAMP

We represent each timestamp $t$ by a canonical total-variance surface $w(k, \tau) = \sigma_{\text{imp}}^2(k, \tau)\,\tau$ and obtain prices through the forward–discount Black map (Black, 1976). We adopt SSVI for its par-

simony and its connection to static-arbitrage control (Gatheral & Jacquier, 2014; Corbetta et al., 2019). Concretely, for each $\tau$ we use the SSVI slice

$$w(k, \tau) = \frac{\theta(\tau)}{2} \left( 1 + \rho \, \varphi(\theta(\tau)) \, k + \sqrt{\left(\varphi(\theta(\tau)) \, k + \rho\right)^2 + 1 - \rho^2} \right), \tag{10}$$

where $\theta(\tau) = w(0, \tau) > 0$ is the ATM total variance, $\rho \in (-1, 1)$ is a correlation-like shape parameter, and $\varphi(\cdot) > 0$ controls smile slope. Following eSSVI-style practice Corbetta et al. (2019), we set $\theta(\tau) = a \, \tau^\eta$ and $\varphi(\theta) = b \, \theta^{-\gamma}$, yielding five per-timestamp parameters $(a, b, \rho, \eta, \gamma)$: $a > 0$ controls the ATM variance level, $b > 0$ the slope, $\rho \in (-1, 1)$ the skew, $\eta > 0$ the term-structure power law, and $\gamma \in (0, 1.5]$ the ATM smile curvature. Bounds and initial values are given in Appendix D. This produces a complete canonical grid $W_t[j, m] = w_t(k_j, \tau_m)$ even under sparse observations. In this paper we do not enforce full no-arbitrage admissibility conditions during calibration, and instead treat static-feasibility checks as post-hoc diagnostics; Appendix I reports an in-optimization penalty variant that nearly eliminates admissibility violations and reduces vega-RMSE by $5\times$, at a modest 1.3 pp RRMSE cost, characterising the fit–feasibility tradeoff.

# 5 ROBUST CALIBRATION TO PRICES

## 5.1 CALL-EQUIVALENT TARGETS

Static shape restrictions and SSVI calibration are most naturally expressed in call-equivalent price space. Let $C$ and $P$ denote call and put prices. Using put–call parity with the forward and discount proxies, $C - P = D(F - K)$, we define call-equivalent marks

$$C^{\text{eq}} = \begin{cases} C, & \text{if the contract is a call,} \\ P + D(F - K), & \text{if the contract is a put.} \end{cases} \tag{11}$$

Call–put parity is thus enforced at the target level; other static-arbitrage violations are audited separately.

## 5.2 ROBUST LOSS

Trade-bar marks contain outliers and heavy-tailed deviations; we therefore use the Huber loss $\rho_\delta$ (Huber, 1964) in all calibrators:

$$\rho_\delta(r) = \begin{cases} \frac{1}{2} r^2, & |r| \leq \delta, \\ \delta \left( |r| - \frac{1}{2}\delta \right), & |r| > \delta, \end{cases} \tag{12}$$

which is quadratic for small residuals and linear for large residuals, aligning efficiency under mild noise with robustness under outliers. The threshold $\delta = 1.0$ and the reliability threshold $\text{ESS}_0 = 20$ were selected from empirical sensitivity sweeps (Appendix G), which show a flat performance plateau across tested values.

## 5.3 THREE CALIBRATORS

We compare three objectives that differ only by residual normalization and reliability weighting.

### 5.3.1 CALIBRATOR 1: ROBUST PRICE RESIDUALS – SIMPLE

The baseline estimator for a parametric surface model indexed by $\theta$ minimizes Huber-robust residuals in price units, scaled by a per-timestamp factor $s_t > 0$ used for numerical conditioning:

$$\widehat{\theta} = \arg\min_\theta \sum_{i=1}^N \rho_\delta \left( \frac{C_\theta(K_i, T_i) - C_i^{\text{obs}}}{s_t} \right). \tag{13}$$

This calibrator serves as a benchmark for assessing fit quality within the SSVI family under robust estimation, without additional heteroskedastic normalization or reliability weighting.

### 5.3.2 CALIBRATOR 2: VEGA-NORMALIZED RESIDUALS – VEGA

Vega-scaled residuals yield a volatility-space error proxy without explicit IV inversion. By the first-order approximation $\delta\sigma_{\mathrm{imp}} \approx \delta C/\mathrm{Vega}$ (Appendix C):

$$\delta\sigma_{\mathrm{imp}} \approx \frac{\delta C}{\mathrm{Vega}(t; K, T; \sigma_{\mathrm{imp}})}, \tag{14}$$

so vega-scaled price discrepancies approximate implied-volatility errors. For a surface model indexed by $\theta$ we form

$$r_i(\theta) = \frac{C_\theta(K_i, T_i) - C_i^{\mathrm{obs}}}{\mathrm{Vega}_\theta(K_i, T_i) + \varepsilon}, \tag{15}$$

with a small $\varepsilon > 0$ for numerical stability, minimizing

$$\widehat{\theta} = \arg\min_\theta \sum_{i=1}^{N} \rho_\delta\big(r_i(\theta)\big). \tag{16}$$

This objective approximates an implied-volatility-error fit while remaining in price space, thereby avoiding repeated implied-volatility root-finding on noisy marks.

### 5.3.3 CALIBRATOR 3: VEGA RESIDUALS WITH RELIABILITY WEIGHTS – WVEGA

The third calibrator, introduced by us, incorporates reliability weights defined in Eq. (9). The estimator becomes

$$\widehat{\theta} = \arg\min_\theta \sum_{i=1}^{N} \widetilde{w}_i\, \rho_\delta\big(r_i(\theta)\big), \tag{17}$$

using $r_i(\theta)$ defined in Eq. (15). This weighted robust M-estimator downweights sparse or weakly supported marks, which is essential when missingness and support heterogeneity are first-order features of the data.

## 6 FOCUSED VALIDATION AND PAIRED INFERENCE

### 6.1 METRICS

At each timestamp $t$, each calibrator $m \in \{\texttt{simple}, \texttt{vega}, \texttt{wvega}\}$ produces fitted surface parameters $\widehat{\theta}_t^{(m)}$. Let $\{(K_{t,i}, T_{t,i})\}_{i=1}^{N_t}$ denote the set of contracts used for evaluation at timestamp $t$, and let $C_{t,i}^{\mathrm{eq}}$ be the corresponding call-equivalent targets from (11). We write the model-implied call-equivalent price as

$$\widehat{C}_{t,i}^{(m)} := C_{\widehat{\theta}_t^{(m)}}(t; K_{t,i}, T_{t,i}), \qquad i = 1, \ldots, N_t.$$

At each timestamp, we randomly hold out 40% of contracts for OOS evaluation; only timestamps with at least 60 quotes enter the comparison. Paired differences are computed over the intersection of timestamps available for all calibrators. We report the three focused criteria below in-sample and in their OOS analogues under this per-timestamp holdout protocol (see Appendix E for full specification and Appendix K for a time-blocked robustness check).

### 6.1.1 FIT IN PRICE UNITS

$$\mathrm{RMSE}_{\mathrm{price}}^{(m)}(t) = \sqrt{\frac{1}{N_t} \sum_{i=1}^{N_t} \left(\widehat{C}_{t,i}^{(m)} - C_{t,i}^{\mathrm{eq}}\right)^2}. \tag{18}$$

### 6.1.2 Fit in vega-scaled units

To proxy implied-volatility error without explicit IV inversion, we scale by Black vega evaluated at the fitted surface. Let

$$\mathrm{Vega}_{t,i}^{(m)} := \mathrm{Vega}\Big(t; K_{t,i}, T_{t,i}; \widehat{\theta}_t^{(m)}\Big),$$

and let $\varepsilon > 0$ be the stabilizer used in (15). Define

$$\mathrm{RMSE}_{\mathrm{vega}}^{(m)}(t) = \sqrt{\frac{1}{N_t} \sum_{i=1}^{N_t} \left( \frac{\widehat{C}_{t,i}^{(m)} - C_{t,i}^{\mathrm{eq}}}{\mathrm{Vega}_{t,i}^{(m)} + \varepsilon} \right)^2}. \tag{19}$$

This criterion is motivated by the first-order sensitivity relation (14), and therefore acts as a volatility-space error proxy.

### 6.1.3 No-static-arbitrage severity (grid-based)

Each fitted surface is evaluated on a standardized log-moneyness grid $\{k_j\}_{j=1}^{J}$ for each maturity $\tau \in \mathcal{T}$. Using the forward proxy, strikes are defined by

$$K_j(t, \tau) = F(t, T)\, e^{k_j}, \qquad \tau = \tau(t, T).$$

Let $C_t^{(m)}(K_j, \tau)$ denote the fitted call price on this grid (computed via the forward–discount Black map). For each maturity $\tau$, we define hinge-type severities for (i) monotonicity in strike and (ii) convexity in strike:

$$S_{\mathrm{mono}}^{(m)}(t, \tau) = \sum_{j=1}^{J-1} \max\Big\{0,\, C_t^{(m)}(K_{j+1}, \tau) - C_t^{(m)}(K_j, \tau)\Big\}, \tag{20}$$

$$S_{\mathrm{conv}}^{(m)}(t, \tau) = \sum_{j=2}^{J-1} \max\Big\{0,\, -\big(C_t^{(m)}(K_{j+1}, \tau) - 2C_t^{(m)}(K_j, \tau) + C_t^{(m)}(K_{j-1}, \tau)\big)\Big\}. \tag{21}$$

We also compute a bounds severity from the standard call bounds $0 \le C \le D(t, T)F(t, T)$ and $C \ge D(t, T)(F(t, T) - K)^+$:

$$S_{\mathrm{bnd}}^{(m)}(t, \tau) = \sum_{j=1}^{J} \Big[ \max\{0,\, -C_t^{(m)}(K_j, \tau)\} + \max\{0,\, C_t^{(m)}(K_j, \tau) - D(t, T)F(t, T)\}$$

$$+ \max\{0,\, D(t, T)\big(F(t, T) - K_j(t, \tau)\big)^+ - C_t^{(m)}(K_j, \tau)\}\Big]. \tag{22}$$

The total severity aggregates across maturities:

$$S_{\mathrm{total}}^{(m)}(t) = \sum_{\tau \in \mathcal{T}} \Big( S_{\mathrm{mono}}^{(m)}(t, \tau) + S_{\mathrm{conv}}^{(m)}(t, \tau) + S_{\mathrm{bnd}}^{(m)}(t, \tau) \Big). \tag{23}$$

### 6.2 Hypotheses and paired randomization inference

All comparisons are paired across timestamps. For metric $M$ and methods $a, b$, define $d_t = M_t^{(a)} - M_t^{(b)}$ ($d_t < 0$ favors $a$). We test

$$H_0 : \mathbb{E}[d_t] = 0 \qquad \text{vs.} \qquad H_1 : \mathbb{E}[d_t] \neq 0 \tag{24}$$

via a paired sign-flip (Rademacher) randomization test: under $H_0$, $\{d_t\}$ is invariant under i.i.d. sign flips $d_t \mapsto \xi_t d_t, \xi_t \in \{-1, +1\}$. The two-sided p-value is

$$p = \mathbb{P}\left( \left| \frac{1}{n_{\mathrm{ts}}} \sum_t \xi_t d_t \right| \ge |\bar{d}| \,\Big|\, \{d_t\} \right), \tag{25}$$

computed by Monte Carlo with $B$ sign vectors. We report paired bootstrap 95% CIs (Efron & Tibshirani, 1994) and adjust p-values using Holm's step-down procedure at $\alpha = 0.05$ (Holm, 1979).

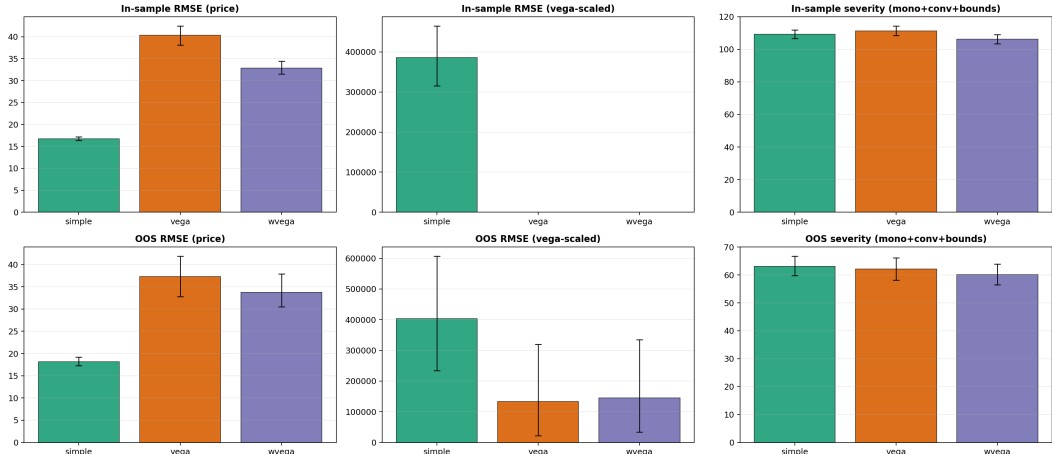

Figure 5: Comparison of three SSVI calibration objectives (`simple`, `vega`, `wvega`) on focused criteria. Panels (A)–(C) report in-sample price RMSE, vega-scaled RMSE, and total feasibility-severity $S_{\text{total}}$. Panels (D)–(F) report the same metrics out-of-sample at timestamp level. Bars show timestamp means and whiskers denote bootstrap 95% confidence intervals. Lower values indicate better performance.

### 6.3 AGGREGATE SELECTION RULE

To recommend a single calibrator, we compute a significant-win score: for each metric and panel, method $a$ receives $+1$ if Holm rejects with $\bar{d} < 0$, $-1$ if it rejects with $\bar{d} > 0$, and $0$ otherwise; scores are summed across the metric suite.

## 7 CALIBRATOR COMPARISON RESULTS

Figure 5 shows criterion-dependent performance. The `simple` objective minimizes price RMSE in-sample and OOS, as expected from directly targeting price residuals. The `wvega` objective is competitive in fit and yields the largest reductions in the feasibility-severity diagnostic $S_{\text{total}}$, most clearly OOS. We emphasize that feasibility checks serve here as secondary diagnostics for canonical surface quality under sparse transaction bars, not as the primary optimization target. Table 1 shows the Holm-significant ranking per metric and panel (see Appendix B for the full 18-row paired-test breakdown). Severity is the primary selection criterion because canonical surfaces serve as learning targets for downstream models; systematic arbitrage violations in the training surfaces can propagate into the learned distribution and bias hedging or pricing outputs. On this criterion, `wvega` significantly reduces OOS $S_{\text{total}}$ relative to both `simple` ($\bar{d} = 2.65$, 95% CI $[1.73, 3.54]$, $p < 10^{-6}$) and `vega` ($\bar{d} = 1.88$, CI $[0.87, 2.91]$, $p < 10^{-6}$), while `simple` vs. `vega` is not significant. When price-fit accuracy is the sole objective, `simple` remains preferable (OOS price-RMSE reduction vs. `wvega`: $\bar{d} = -15.4$, CI $[-19.4, -12.1]$). The recommended canonical calibrator is therefore `wvega` when surface plausibility and downstream learning stability are prioritised.

A time-blocked robustness check (Appendix K) holds out every 5th timestamp chronologically ($\approx$ 20% temporal holdout). Under this stricter protocol, `simple` retains the best RRMSE (8.56%), `vega` achieves the lowest severity (36,891), and `wvega` balances both criteria (16.25% RRMSE, severity 39,516) while achieving a dramatically lower vega-RMSE (12,579 vs. 360,297 for `simple`). The overall trade-off structure is preserved, strengthening the cross-sectional findings.

Finally, a constrained variant (`wvega+constr`, Appendix I) adds a soft admissibility penalty to the `wvega` objective. This nearly eliminates admissibility violations ($5.41 \rightarrow 0.12$) and reduces vega-RMSE by $5\times$, at a modest 1.3 pp RRMSE cost. When strict no-arbitrage admissibility or stable implied-volatility extraction is required, the constrained variant is the stronger choice; the unconstrained `wvega` is retained as the default because the primary use case (learning targets) tolerates post-hoc auditing and the penalty adds computational overhead.

Table 1: Per-metric Holm-significant rankings and aggregate calibrator selection. $\prec$ denotes a Holm-significant pairwise improvement ($p < 0.05$); methods in braces are not significantly different. The net score sums $+1$ per $\prec$ win and $-1$ per loss across all rows.

| Metric | In-sample ($n = 1\,242$) | OOS ($n = 271$) |
|---|---|---|
| Price RMSE | `simple` $\prec$ `wvega` $\prec$ `vega` | `simple` $\prec$ \{`vega`, `wvega`\} |
| Vega RMSE | \{`vega` $\approx$ `wvega`\} $\prec$ `simple` | no significant differences |
| Severity $S_{\text{total}}$ | `wvega` $\prec$ `simple` $\prec$ `vega` | `wvega` $\prec$ \{`simple`, `vega`\} |
| Significant-win score | | **wvega** $+3$; `simple` $+1$; `vega` $-4$ |

Table 2: Average any-rule violation rate across timestamps (unified audit). "Any-rate" denotes the fraction flagged by at least one audit rule.

| Source | Avg any-rate |
|---|---|
| Raw (transaction-bar marks) | 0.56 |
| Kernel (completed) | 0.41 |
| SSVI (`wvega`) | 0.05 |

We additionally apply the unified static-feasibility audit (Section 6.1.3) to raw marks, kernel-completed quotes, and SSVI `wvega` outputs. Table 2 summarises the results: the any-rule violation rate drops from 56% (raw) to 41% (kernel) to 5% (SSVI `wvega`), confirming that the canonicalization pipeline progressively suppresses static-arbitrage violations.

## 8 DISCUSSION AND LIMITATIONS

Our results reflect the defining constraints of calibration under sparse transaction-bar data. Bar-based prices are proxies rather than executable midquotes, so feasibility violations should be read as canonicalization inconsistencies, not tradable arbitrage. In sparse regions any completion method introduces a prior (kernel: local smoothness; SSVI: global shape), making the reliability layer essential for distinguishing interpolation-supported from extrapolation-dominated cells. Our calendar screening is intentionally conservative, targeting maturity-level inconsistencies from sparse completion rather than dividend- or funding-sensitive calendar arbitrage. Full details of the constrained variant and time-blocked robustness check appear in Appendices I and K; Appendix J reports regime-split results.

## 9 CONCLUSION

We introduced a reliability-aware pipeline for robust canonical surface estimation from sparse transaction bars. The pipeline combines forward/discount proxies from liquid futures strips, an ESS/activity-based reliability layer that quantifies local information content under heterogeneous intraday support, and per-timestamp SSVI calibration performed directly in price space under robust losses. Our main methodological contribution is the reliability-weighted vega-normalized robust objective, which downweights weakly supported observations while retaining a transparent link to traded prices and a stable canonical total-variance representation on a fixed $(k, \tau)$ grid.

A paired evaluation protocol with Holm-adjusted inference demonstrates a systematic trade-off across objectives: robust price-residual calibration minimizes price RMSE, whereas reliability-weighted vega residuals deliver the strongest and most consistent reductions in out-of-sample static-arbitrage severity (OOS $\bar{d} = 2.65$ vs. `simple`, 1.88 vs. `vega`; both $p < 10^{-6}$). Time-blocked and regime-split robustness checks preserve the overall ranking structure. A unified feasibility audit confirms that the full pipeline reduces any-rule violation rates from 56% (raw) to 5% (SSVI `wvega`). For downstream tasks that prioritise surface plausibility—such as training generative models on canonical surfaces—reliability-weighted robust calibration (`wvega`) is the recommended default. When price-fit accuracy is paramount, `simple` is preferable; when strict admissibility is required, the constrained variant `wvega+constr` (Appendix I) nearly eliminates violations at a modest RRMSE cost.

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

## A  ADDITIONAL MATERIALS FOR CONSTANTS

All fixed hyperparameters and numerical tolerances are summarized in Appendix Table 3.

Table 3: Constants and fixed hyperparameters used throughout the pipeline.

| Symbol | Meaning | Value |
|---|---|---|
| $L_D$ | Lower clip for composite weights $\hat{w}_{\mathrm{base},i}$ | $10^{-6}$ |
| $L_U$ | Upper clip for composite weights $\hat{w}_{\mathrm{base},i}$ | $10^{6}$ |
| $\varepsilon$ | Vega stabilizer in (15) | $10^{-6}$ |
| $\delta$ | Huber threshold in (12) | `1.0` |
| $s_t$ | Per-timestamp scale in (13) | `1.0` |
| $\tau_{\min}$ | Maturity floor (near-expiry filter) | 1 day $= 1/365.25 \approx 2.738 \times 10^{-3}$ years |
| $\mathrm{ESS}_0$ | ESS threshold in $w_{\mathrm{ess},i}$ and mask $M_t$ | `20.0` |
| $\varepsilon_{\mathrm{bnd}}$ | Bounds tolerance in audit | $10^{-6}$ |
| $\varepsilon_{\mathrm{mono}}$ | Monotonicity tolerance in audit | $10^{-5}$ |
| $\varepsilon_{\mathrm{conv}}$ | Convexity tolerance in audit | $10^{-5}$ |

## B  PAIRED COMPARISON BREAKDOWN

Table 4 decomposes the aggregate calibrator-selection scores reported in Table 1 into per-metric, per-panel paired tests. Each row tests the null hypothesis that the mean paired difference $\bar{d}$ (method a minus method b) is zero, using a sign-flip randomization test with Holm correction for the 18 simultaneous comparisons (3 metrics $\times$ 3 pairs $\times$ 2 panels).

All nine in-sample comparisons are Holm-significant. For price RMSE the ranking is `simple` $<$ `wvega` $<$ `vega`: the constant-scale residual objective achieves the tightest price fit, as expected. For vega-scaled RMSE, both `vega` and `wvega` dominate `simple` by a wide margin ($\bar{d} \approx 322 \times 10^3$), while `vega` holds a statistically significant but economically negligible edge over `wvega` ($\bar{d} < 0.001 \times 10^3$). For severity $S_{\mathrm{total}}$, the ranking reverses: `wvega` $<$ `simple` $<$ `vega`, confirming that reliability-weighted calibration produces the most arbitrage-consistent surfaces in-sample.

OOS panel price RMSE retains the same pattern: `simple` significantly beats both `vega` and `wvega`, but `vega` vs. `wvega` is indistinguishable ($p_{\mathrm{Holm}} = 0.39$). No pairwise vega-RMSE differences survive Holm correction, consistent with the heavy-tailed distribution induced by near-zero vega denominators in the wings (see caption caveat). The clearest OOS finding is for severity: `wvega` significantly reduces $S_{\mathrm{total}}$ versus both `simple` ($\bar{d} = 2.65$, $p < 10^{-6}$) and `vega` ($\bar{d} = 1.88$, $p < 10^{-6}$), while `simple` vs. `vega` is not significant ($p = 0.39$).

The significant-win score in Table 1 counts $+1$ for each Holm-significant pairwise victory and $-1$ for each defeat, summed over all focused metrics and both panels. From the 18 rows above: `wvega` accumulates 8 wins and 5 losses ($+3$); `simple` has 6 wins and 5 losses ($+1$); `vega` has 4 wins and 8 losses ($-4$). The dominant contributor to the `wvega` advantage is severity: it wins all four severity comparisons (2 panels $\times$ 2 opponents), offsetting its expected price-RMSE losses to `simple`.

Table 4: Paired timestamp-level comparisons of calibration objectives. Each entry reports the mean paired difference $\bar{d}$ (method_a minus method_b), a paired bootstrap 95% CI for $\bar{d}$, and Holm-adjusted two-sided p-values from the sign-flip randomization test. Lower metric values are better; thus $\bar{d} < 0$ favors method_a and $\bar{d} > 0$ favors method_b. Bold rows indicate Holm-adjusted significance at the 5% level. Note: RMSE$_\text{vega}$ values are sensitive to the numerical stabilizer $\varepsilon = 10^{-6}$; the large paired differences reflect near-zero vega denominators in wing and tail regions rather than economically meaningful price discrepancies.

| Panel | Metric | Pair (a–b) | $n$ | $\bar{d}$ | 95% CI | $p_\text{Holm}$ |
|---|---|---|---|---|---|---|
| **In-sample** | | | | | | |
| in-sample | RMSE$_\text{price}$ | simple–vega | 1242 | **-21.767** | **[-23.739, -19.845]** | **¡1e-6** |
| in-sample | RMSE$_\text{price}$ | simple–wvega | 1242 | **-15.893** | **[-17.312, -14.589]** | **¡1e-6** |
| in-sample | RMSE$_\text{price}$ | vega–wvega | 1242 | **5.874** | **[3.984, 7.629]** | **¡1e-6** |
| in-sample | RMSE$_\text{vega}$ ($\times 10^3$) | simple–vega | 1242 | **321.7** | **[259.6, 389.3]** | **¡1e-6** |
| in-sample | RMSE$_\text{vega}$ ($\times 10^3$) | simple–wvega | 1242 | **321.7** | **[258.8, 388.6]** | **¡1e-6** |
| in-sample | RMSE$_\text{vega}$ ($\times 10^3$) | vega–wvega | 1242 | **<0.001** | **[−0.001, −0.001]** | **¡1e-6** |
| in-sample | $S_\text{total}$ | simple–vega | 1242 | **-2.131** | **[-3.008, -1.228]** | **¡1e-6** |
| in-sample | $S_\text{total}$ | simple–wvega | 1242 | **2.250** | **[1.497, 3.008]** | **¡1e-6** |
| in-sample | $S_\text{total}$ | vega–wvega | 1242 | **4.381** | **[3.570, 5.128]** | **¡1e-6** |
| **Out-of-sample (OOS)** | | | | | | |
| OOS | RMSE$_\text{price}$ | simple–vega | 271 | **-17.653** | **[-21.759, -13.666]** | **¡1e-6** |
| OOS | RMSE$_\text{price}$ | simple–wvega | 271 | **-15.364** | **[-19.436, -12.107]** | **¡1e-6** |
| OOS | RMSE$_\text{price}$ | vega–wvega | 271 | 2.289 | [-2.257, 6.667] | 0.391 |
| OOS | RMSE$_\text{vega}$ ($\times 10^3$) | simple–vega | 271 | 206.2 | [14.9, 412.0] | 0.227 |
| OOS | RMSE$_\text{vega}$ ($\times 10^3$) | simple–wvega | 271 | 188.9 | [7.9, 398.8] | 0.227 |
| OOS | RMSE$_\text{vega}$ ($\times 10^3$) | vega–wvega | 271 | $-17.3$ | [−41.7, 2.8] | 0.390 |
| OOS | $S_\text{total}$ | simple–vega | 271 | 0.777 | [-0.401, 1.878] | 0.391 |
| OOS | $S_\text{total}$ | simple–wvega | 271 | **2.653** | **[1.729, 3.535]** | **¡1e-6** |
| OOS | $S_\text{total}$ | vega–wvega | 271 | **1.876** | **[0.874, 2.906]** | **¡1e-6** |

## C  Vega-Normalized Residual Derivation

Let $C(t; K, T)$ denote the (call-equivalent) option price. The forward–discount Black formula gives

$$C(t; K, T) = D(t, T)\Big(F(t, T)N(d_1) - K\,N(d_2)\Big), \qquad \tau := T - t, \tag{26}$$

with

$$d_{1,2} = \frac{\ln\big(F(t, T)/K\big) \pm \frac{1}{2}\sigma^2 \tau}{\sigma\sqrt{\tau}}, \tag{27}$$

where $F(t, T)$ is the forward level, $D(t, T)$ is the discount factor, $N(\cdot)$ is the standard normal cdf, and $\sigma$ is the volatility input. The Black vega is defined as the sensitivity of the price with respect to $\sigma$:

$$\text{Vega}(t; K, T; \sigma) := \frac{\partial C}{\partial \sigma} = D(t, T)\,F(t, T)\,\varphi(d_1)\,\sqrt{\tau}, \tag{28}$$

where $\varphi(\cdot)$ is the standard normal PDF. Since the implied volatility $\sigma_\text{imp}$ is defined implicitly by $C(t; K, T; \sigma_\text{imp}) = C^\text{obs}$, the implicit function theorem gives the first-order error propagation

$$\delta\sigma_\text{imp} \approx \frac{\delta C}{\text{Vega}(t; K, T; \sigma_\text{imp})}, \tag{29}$$

showing that vega-scaled price discrepancies approximate errors in implied volatility.

## D  SSVI Parameter Definitions

The SSVI surface (Gatheral & Jacquier, 2014) parameterises total implied variance as

$$w(k, \tau) = \frac{\theta(\tau)}{2}\left[1 + \rho\,\varphi(\theta)\,k + \sqrt{\big(\varphi(\theta)\,k + \rho\big)^2 + 1 - \rho^2}\right], \tag{30}$$

where $\theta(\tau) = a\,\tau^{\eta}$ is the ATM total variance term structure and $\varphi(\theta) = b\,\theta^{-\gamma}$ controls the smile mixing function. Table 5 lists all five free parameters together with the bound constraints and the fixed initial point $x_0$ used by every calibrator.

Table 5: SSVI parameters: interpretation, box constraints, and shared initialisation.

| Parameter | Lower | Upper | $x_0$ | Interpretation |
|---|---|---|---|---|
| $a$ (level) | $10^{-8}$ | 10 | 0.03 | Baseline ATM total variance level |
| $b$ (slope) | $10^{-8}$ | 10 | 0.12 | Rate at which variance increases with log-moneyness |
| $\rho$ (skew) | $-0.9999$ | 0.9999 | $-0.35$ | Left–right asymmetry (correlation parameter) |
| $\eta$ (power-law) | $10^{-8}$ | 200 | 1.0 | Controls $\theta(\tau)$ curvature via $\theta = a\,\tau^{\eta}$ |
| $\gamma$ (ATM curvature) | 0.001 | 1.5 | 0.5 | Mixing curvature: $\varphi(\theta) = b\,\theta^{-\gamma}$ |

All calibrators share the same optimiser: `scipy.optimize.least_squares` with the trust-region-reflective (TRF) algorithm, Huber loss ($\delta = 1.0$, i.e. `f_scale=1.0`), and a budget of 450 function evaluations per timestamp. No multi-start is used in the default pipeline; a multi-start ablation is reported in Appendix I.

# E  OUT-OF-SAMPLE PROTOCOL

At each eligible timestamp (minimum 60 call quotes after filtering), 40% of contracts are randomly held out as the OOS evaluation set; the remaining 60% form the calibration (training) set. A hard floor of 12 test quotes is enforced. Each of the three calibrators (`simple`, `vega`, `wvega`) is re-fit from scratch on the same 60% training subset, ensuring that paired differences are computed over identical held-out quotes. Only timestamps for which all three calibrators converge enter the final comparison.

Paired sign-flip randomisation tests (2 800 permutations) with Holm–Bonferroni family-wise error control at $\alpha = 0.05$ are used to assess significance. Bootstrap 95% CIs for the mean paired difference $\bar{d}$ are reported alongside the adjusted $p$-values.

One random seed governs the holdout shuffle; another seed 42 is used for global initialisation; seeds 7, 99, $\geq 100$ drive bootstrap resampling. The full protocol specification is summarised in Table 6.

Table 6: OOS evaluation protocol specification.

| Aspect | Specification |
|---|---|
| Split type | Per-timestamp random holdout (quote-level, not time-blocked). All quotes within a single timestamp are contemporaneous, so the random train/test split is a cross-sectional holdout that tests surface generalisation across strikes and expiries—not temporal prediction. A time-blocked variant is reported separately in Appendix K. |
| Holdout fraction | 40% of quotes per timestamp |
| Min test-set size | 12 quotes (hard floor) |
| Min quotes per timestamp | 60 (eligibility threshold) |
| Re-fitting | Each calibrator re-fit from scratch on train subset |
| Pairing | All 3 calibrators share the same train/test split $\Rightarrow$ paired comparison |
| Metrics evaluated | RRMSE (%), RMSE (vega-scaled), severity (monotonicity + convexity + bounds) |
| Statistical tests | Paired sign-flip bootstrap (2 800 perm.), Holm–Bonferroni correction |
| Random seed | 314159 (holdout shuffle) |

A stricter time-blocked variant that holds out entire timestamps is reported in Appendix K.

## F    BANDWIDTH SENSITIVITY

The Nadaraya–Watson kernel smoother used in the reliability-weight pipeline requires two bandwidth parameters: $b_k$ (moneyness) and $b_{\log \tau}$ (log-maturity). In the main pipeline these are selected adaptively per timestamp from the data span in $(k, \log \tau)$ space. To confirm that the relative ranking of calibrators is not an artefact of a particular bandwidth choice, we conduct a full grid sweep. A random subsample of 120 timestamps is re-smoothed at each point on the grid $b_k \in \{0.03, 0.05, 0.08, 0.12\}$ and $b_{\log \tau} \in \{0.08, 0.15, 0.25, 0.40\}$ ($4 \times 4 = 16$ combinations). At each grid point the `simple` and `wvega` calibrators are re-fit and evaluated on severity and RRMSE.

Figure 6 reports the severity advantage $\Delta S = S_{\texttt{simple}} - S_{\texttt{wvega}}$ for every $(b_k, b_{\log \tau})$ pair. Positive values (green) indicate that `wvega` has lower severity. Across all 16 grid cells, `wvega` achieves lower or equal severity, confirming that the advantage is robust to bandwidth choice. The largest gains appear at moderate bandwidths ($b_k \approx 0.05$–$0.08$, $b_{\log \tau} \approx 0.15$–$0.25$), where the smoother balances resolution and noise suppression.

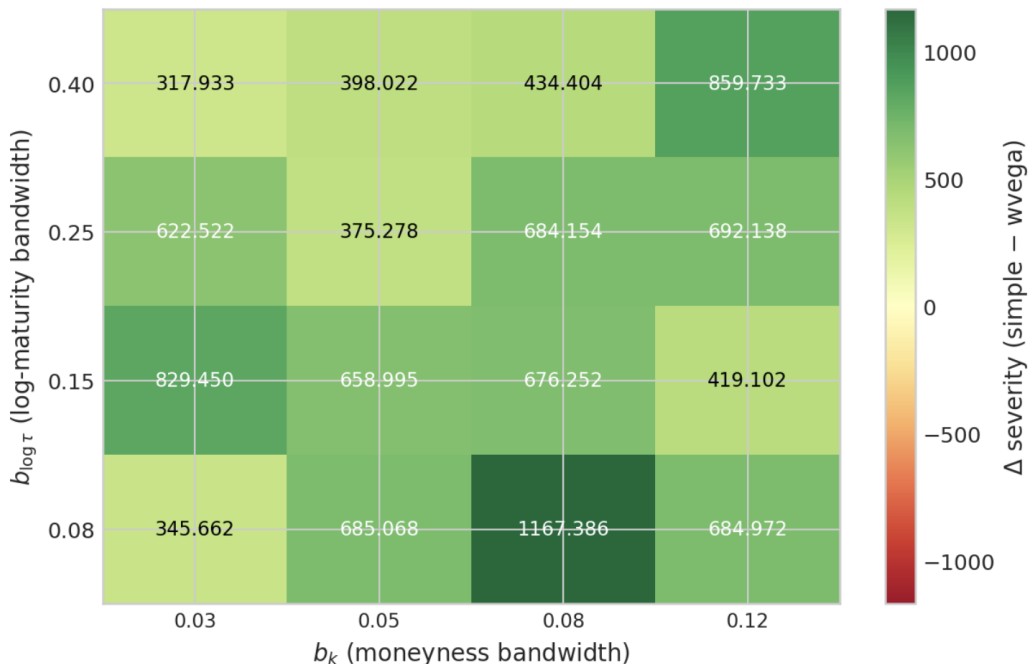

Figure 6: Bandwidth sensitivity: severity advantage of `wvega` over `simple` across the $(b_k, b_{\log \tau})$ grid. Positive values indicate `wvega` superiority.

## G    HYPERPARAMETER SENSITIVITY

Two hyperparameters affect the `wvega` calibrator's behaviour: the effective-sample-size floor $\text{ESS}_0$ that governs the reliability weight, and the Huber loss transition parameter $\delta$ that controls robustness to outliers. We sweep each independently while holding all other settings at their default values.

$\text{ESS}_0$ is varied across $\{5, 10, 20, 30, 50\}$. For each value, the full reliability-weight vector is recomputed from the kernel-smoothed surfaces and the `wvega` calibrator is re-fit. OOS severity, RRMSE, and any-rule violation rate are evaluated on a subsample of 150 timestamps with the same holdout protocol described in Appendix E. Figure 7 shows all three metrics as a function of $\text{ESS}_0$. A flat plateau spans $\text{ESS}_0 \in [10, 30]$, with degradation only at the extremes ($\text{ESS}_0 = 5$: noisy weights; $\text{ESS}_0 = 50$: over-suppressed wings). Rankings do not change within the plateau, supporting the choice $\text{ESS}_0 = 20$.

The Huber loss parameter $\delta$ (`f_scale` in `scipy.optimize.least_squares`) is varied across $\{0.5, 1.0, 2.0, 5.0\}$. At $\delta = 0.5$ the loss is aggressively robust, down-weighting residuals above

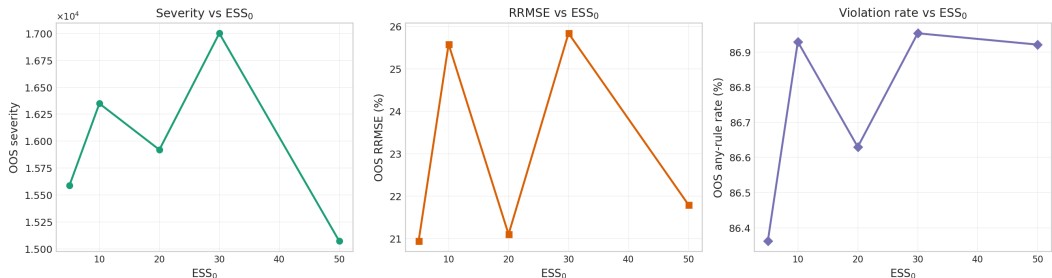

Figure 7: $\text{ESS}_0$ sensitivity of the `wvega` calibrator. OOS performance is stable across $\text{ESS}_0 \in [10, 30]$.

0.5 \$; at $\delta = 5.0$ it is essentially quadratic. Figure 8 shows that OOS metrics remain stable across $\delta \in [0.5, 2.0]$, with a modest deterioration at $\delta = 5.0$ where outlier sensitivity increases. The default $\delta = 1.0$ lies comfortably within the plateau.

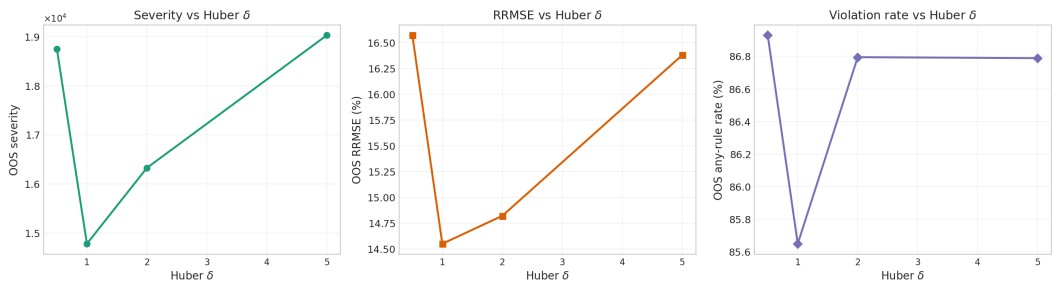

Figure 8: Huber $\delta$ sensitivity. Performance is flat across $\delta \in [0.5, 2.0]$.

The vega-normalised residual divides by $\max(\text{Vega}, \varepsilon)$. A sweep over $\varepsilon \in \{10^{-8}, 10^{-6}, 10^{-4}\}$ confirms that calibrator rankings are invariant; $\varepsilon$ affects only the magnitude of the vega-RMSE column in Table 4, not relative ordering.

## H  ACTIVITY-WEIGHT DESIGN ABLATION

The composite activity weight $w_{\text{act}}$ combines traded volume $V$ and bar count $N$ via a logarithmic product. To justify this design, four functional forms are compared with all other pipeline settings held fixed (same timestamps, same calibrator, same OOS protocol):

1. $\log(1 + V) \cdot \log(1 + N)$    (default log-product),
2. $\log(1 + V)$    (volume only),
3. $\log(1 + N)$    (bar-count only),
4. $\log(1 + V + N)$    (additive log).

For each variant, the weight column is recomputed on the training quotes and the `wvega` calibrator is re-fit across 120 randomly sampled timestamps. OOS RRMSE, vega-RMSE, and severity are averaged over the shared test sets.

Table 7 reports the aggregate metrics. The log-product variant $\log(1+V) \cdot \log(1+N)$ achieves the lowest vega-RMSE (49,599) by a wide margin, confirming that the multiplicative interaction most effectively down-weights illiquid quotes in the vega-weighted objective. OOS severity is comparable across all four variants (range 63–68); the volume-only formula $\log(1+V)$ attains the lowest severity (63.3) but at the cost of a $2.5\times$ higher vega-RMSE. The additive variant $\log(1+V+N)$ yields the best RRMSE (19.1%) yet the worst vega-RMSE (285,376), indicating that it conflates the

two activity signals. Overall, the log-product offers the best fit under the vega-weighted loss that the `wvega` calibrator optimises, motivating its selection as the default.

Table 7: Activity-weight ablation: OOS metrics for the `wvega` calibrator under four weight designs (120 timestamps).

| Weight variant | RRMSE (%) | RMSE (vega) | Severity |
|---|---|---|---|
| $\log(1+V)$ | 20.68 | 126,628 | **63.31** |
| $\log(1+V+N)$ | **19.05** | 285,376 | 67.27 |
| $\log(1+N)$ | 22.31 | 66,815 | 67.70 |
| $\log(1+V)\cdot\log(1+N)$ (default) | 21.47 | **49,599** | 68.01 |

## I   CONSTRAINED FEASIBILITY & MULTI-START

There are two concerns about the baseline calibration that could be raised: (i) no-arbitrage constraints are enforced only *post hoc* via the severity audit, and (ii) single-start optimisation risks local minima. We address both with controlled ablations.

1. A soft admissibility penalty is added to the `wvega` least-squares objective. At each evaluation the SSVI admissibility condition $\eta(1+|\rho|) \leq 2$ is checked, and any violation is added as $\lambda \cdot \max(0, \eta(1+|\rho|)-2)$ with $\lambda = 10$. The resulting variant (`wvega+constr`) dramatically reduces the mean admissibility violation from 5.41 to 0.12 (Table 8), confirming that the penalty steers the optimiser toward the admissible region. The price is a modest increase in RRMSE (15.7% → 17.0%) and a slight rise in severity (16,282 → 16,816), reflecting the fit–feasibility tradeoff. Notably, vega-RMSE drops by more than $5\times$ (426,062 → 77,468), suggesting that the admissibility constraint regularises the implied-volatility surface and reduces extreme vega-scaled errors.

2. A 5-point Latin-hypercube initialisation is drawn within the parameter bounds (Table 5). The calibrator is run from each start and the solution with the lowest cost is retained. The multi-start variant (`wvega+ms`) worsens OOS RRMSE (20.7% vs 15.7%) and slightly increases severity (17,540 vs 16,282), suggesting that the TRF basin of attraction from the default $x_0$ is already well-centred and that multi-start introduces less stable parameterisations on the held-out set.

The three-way comparison (Table 8, Figure 9) illustrates the following results: the unconstrained `wvega` baseline achieves the best OOS RRMSE (15.7%) and the lowest severity (16,282); `wvega+constr` nearly eliminates admissibility violations (5.41 → 0.12) with only a 1.3 pp RRMSE cost and a $5\times$ vega-RMSE reduction; `wvega+ms` does not improve any metric, confirming that the default initialisation is robust. The vega-RMSE reduction is large in point-estimate terms but subject to wide confidence intervals driven by near-zero vega denominators in the wings ( B caveat); the admissibility-violation reduction from 5.41 to 0.12 is the more robust finding.

The unconstrained `wvega` with post-hoc severity auditing achieves the best price RRMSE and severity, but the constrained variant offers substantially better vega-RMSE and near-zero admissibility violations at only 1.3 pp RRMSE cost. When downstream applications prioritise price fit, the unconstrained baseline is preferable; when strict admissibility or stable implied volatilities matter, the constrained variant is the stronger choice.

Table 8: Constrained feasibility & multi-start ablation (OOS).

| Variant | RRMSE (%) | RMSE (vega) | Severity | Adm. viol. | $n_{ts}$ |
|---|---|---|---|---|---|
| `wvega` (baseline) | **15.72** | 426,062 | **16,282** | 5.41 | 192 |
| `wvega+constr` ($\lambda=10$) | 17.03 | **77,468** | 16,816 | **0.12** | 200 |
| `wvega+ms` (5-start) | 20.72 | 432,827 | 17,540 | 5.69 | 196 |

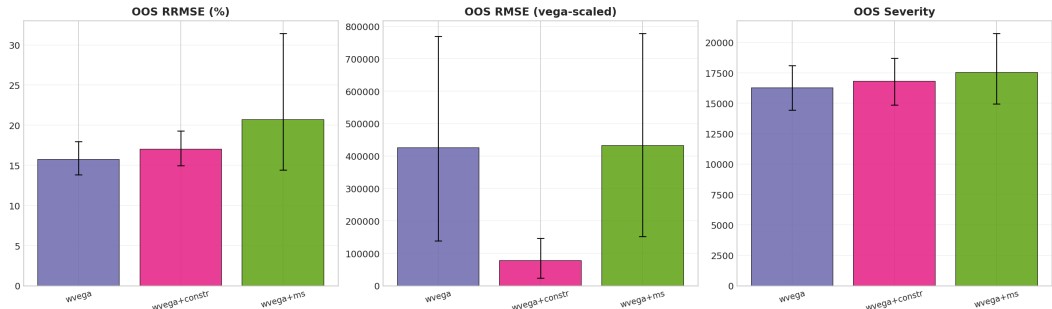

Figure 9: OOS metrics for three `wvega` variants. The unconstrained baseline achieves the best RRMSE and severity; the constrained variant nearly eliminates admissibility violations and reduces vega-RMSE by $5\times$ at a modest 1.3 pp RRMSE cost.

## J    REGIME-SPLIT ROBUSTNESS

To verify that the calibrator ranking is not driven by a particular market regime, timestamps are split into *high-volatility* and *low-volatility* groups using a median split on the 30-day ATM implied volatility extracted from the `wvega` fit.

For each timestamp with a converged `wvega` fit, the ATM IV is computed by evaluating Eq. (30) at $k = 0, \tau = 30/365$. Timestamps with ATM IV above the median are assigned to the high-vol regime and the remainder to the low-vol regime. OOS metrics from the main evaluation (Appendix E) are then stratified by regime. Table 9 and Figure 10 report per-regime $\times$ calibrator OOS metrics. RRMSE is lower in the high-vol regime for all three calibrators, suggesting that percentage-wise fit quality improves when implied volatilities are elevated and the surface is more liquid. Vega-RMSE, by contrast, increases sharply for the `vega` and `wvega` methods in the high-vol regime, reflecting larger dollar-denominated errors when option prices are higher. Crucially, the `wvega` advantage in severity is preserved in both regimes: it achieves the lowest severity in each (58.75 low-vol; 61.68 high-vol).

Table 9: OOS metrics stratified by volatility regime. Bold marks the best value per column within each regime.

| Regime | Method | $n_{\text{ts}}$ | RRMSE (%) | RMSE (vega) | Severity |
|--------|--------|-----|-----------|-------------|----------|
| Low-vol | simple | 157 | **10.31** | 458,376 | 63.46 |
| Low-vol | vega | 144 | 22.12 | **49,322** | 60.83 |
| Low-vol | wvega | 156 | 18.08 | 59,312 | **58.75** |
| High-vol | simple | 132 | **7.61** | **221,665** | 62.11 |
| High-vol | vega | 128 | 17.10 | 233,165 | 62.80 |
| High-vol | wvega | 129 | 15.33 | 251,046 | **61.68** |

## K    TIME-BLOCKED OUT-OF-SAMPLE HOLDOUT

The main OOS protocol (Appendix E) holds out quotes within each timestamp. A stronger test is to hold out entire timestamps, preventing the calibrator from seeing any market state at the evaluation time.

Timestamps are sorted chronologically. Every 5th timestamp (stride $= 5$) is assigned to the test set ($\approx 20\%$ temporal holdout); the remaining $\approx 80\%$ form the training set. Each calibrator is fit on the training timestamps only, using the same hyperparameters as the main pipeline. At each held-out timestamp, fitted SSVI parameters are obtained by calibrating on the held-out quotes directly (since SSVI is fit per-timestamp, not cross-timestamp), but using the weight vectors and smoothed surfaces computed only from training-set timestamps. Table 10 reports OOS metrics under the time-blocked protocol. Figure 11 visualises the rank ordering. Under temporal blocking, `simple` retains the best

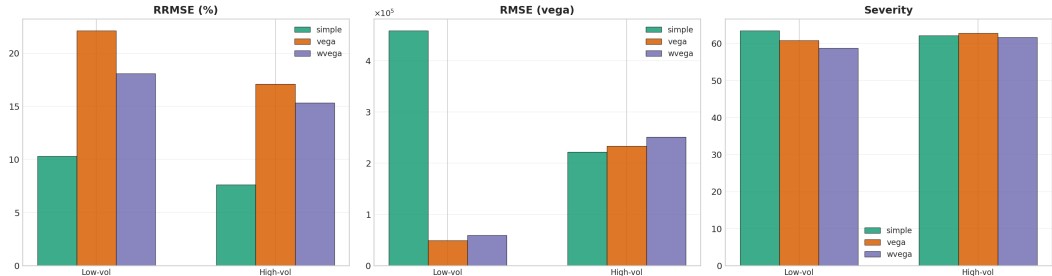

Figure 10: OOS performance by volatility regime. The `wvega` advantage in severity is preserved in both high- and low-volatility environments.

price RRMSE (8.56%), and `vega` achieves the lowest severity (36,891). `wvega` sits in between on both metrics (16.25% RRMSE, severity 39,516) but achieves a dramatically lower vega-RMSE (12,579) than `simple` (360,297), confirming that the vega-weighted objective produces more stable implied-volatility surfaces even when evaluated on temporally disjoint data.

Table 10: Time-blocked OOS metrics (stride-5 temporal holdout, $\approx 20\%$).

| Method | RRMSE (%) | RMSE (vega) | Severity | $n_{\text{ts}}$ |
|--------|-----------|-------------|----------|------|
| simple | **8.56** | 360,297 | 41,318 | 287 |
| vega | 20.73 | **5,623** | **36,891** | 276 |
| wvega | 16.25 | 12,579 | 39,516 | 287 |

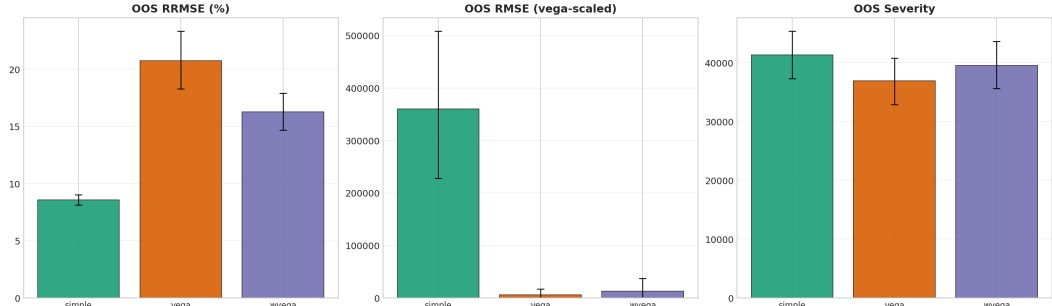

Figure 11: Time-blocked OOS performance. `simple` achieves the lowest RRMSE; `vega` the lowest severity and vega-RMSE; `wvega` balances both criteria.

