# OpenReview forum: "Calibration under Sparse Data: Robust Canonical Surface Estimation from Transaction Bars"
_mathai.club/MathAI/2026/Conference — 2026 Oral_

### Official Review · Reviewer_PZRt · 2026-03-11
**Reliability-Aware SSVI Canonicalization for Sparse Intraday Option Panels A Rigorous Pipeline with Incremental Novelty**

**Rating:** 6
**Confidence:** 4

**Review:**

Summary:
This paper proposes a canonicalization pipeline for implied volatility (IV) surfaces estimated from sparse intraday transaction bars. The core contributions are: (i) an ESS-based reliability layer using kernel receptive fields, (ii) a reliability-weighted vega-normalized robust calibration objective (wvega), and (iii) a unified static-arbitrage audit with hinge-type severity measures. The recommended calibrator reduces OOS arbitrage severity substantially (violation rate dropping from 0.56 on raw marks to 0.05 under SSVI-wvega).

Strengths:

1) Well-motivated problem: The mismatch between quote-centric IV surface assumptions and real-world sparse transaction bars is genuine and underappreciated. The reliability framing is the right lens.

2) Methodological care: The paired randomization inference with Holm correction is statistically principled. Reporting both violation rates and hinge severities is a meaningful improvement over incidence-only audits.

3) Strong empirical outcome: The 91% reduction in any-rule violation rate (0.56 → 0.05) is compelling and practically significant for downstream ML tasks.

4) Honest scope: The authors clearly acknowledge that no-arbitrage constraints are not enforced in-optimization, and that bar prices are proxies appropriate epistemic humility.

Weaknesses
1. Novelty is incremental. SSVI is well-established (Gatheral & Jacquier, 2014; Corbetta et al., 2019). Huber-robust calibration and vega-normalized residuals are standard. The primary novel element reliability weighting via ESS is a natural extension. The contribution is sound engineering, but the theoretical novelty is modest for a MathAI venue.
2. No-arbitrage constraints are post-hoc only. The optimizer does not enforce admissibility conditions. For a paper whose main evaluation criterion is static-arbitrage reduction, enforcing constraints during calibration (or at least comparing against a constrained variant) is an important missing baseline.
3. Bandwidth selection is unaddressed. The ESS diagnostic in Equations (1)–(3) is sensitive to the choice of bandwidths (bk,blog⁡τ)(b_k, b_{\log\tau})
(bk​,blogτ​), yet no principled selection method, sensitivity analysis, or ablation over bandwidth is provided. This is a material gap.
4. Single asset, single regime. All results are on SPX options over ~5 months (Oct 2025–Mar 2026). Generalization to other underlyings, volatility regimes, or crisis periods is not established.
5. Table 1 is under-reported. The win-score (+3, +1, −4) summarizing calibrator selection is stated without showing the underlying per-metric breakdown, making it difficult to assess which metrics drive the recommendation.

---

> ### Author Rebuttal · Authors · 2026-03-13
>
> Thank you for the careful and constructive review. We appreciate your recognition of the practical relevance of the problem, the statistical rigor of the evaluation, and the strength of the empirical feasibility improvements!
>
> We agree that the novelty is incremental at the level of individual components. Our contribution is in the integrated reliability-aware pipeline for sparse intraday transaction-bar data, a setting that differs materially from the quote-based panels more commonly studied. In particular, ESS/activity information enters the calibration objective itself, not only the diagnostics, and the method is designed for robust price-space estimation under uneven local support.
>
> We also agree that a constrained no-arbitrage baseline would strengthen the paper. Our current use of post-hoc auditing was intentional, since we wanted to isolate the effect of the calibration objective alone, but we will make this choice and its limitations much clearer.
>
> The comments on bandwidth selection, limited empirical scope, and the compressed reporting in Table 1 are all well taken. We will address these by adding a bandwidth sensitivity analysis, clarifying that the current results are a benchmark-market proof of concept rather than a universal claim, and expanding the per-metric breakdown behind the aggregate selection score.
>
> We thank the reviewer again for the balanced assessment and helpful suggestions!

---

### Official Review · Reviewer_PJyJ · 2026-03-11
**Sound methodology undermined by citation errors, missing specifications, and reproducibility gaps**

**Rating:** 3
**Confidence:** 3

**Review:**

Summary
This paper presents a method for calibrating implied volatility surfaces from sparsely observed options transaction data using weighted kernel smoothing with SSVI parametrization. The approach incorporates activity-based weighting, Huber-loss robust estimation, and a paired sign-flip randomization test with Holm correction for evaluation.
Strengths

The problem setting (sparse transaction bar data) is practically relevant and underserved in the literature.
SSVI is an appropriate parametric framework for implied volatility surfaces.
The statistical evaluation methodology (paired sign-flip test with Holm correction) is sound and well-chosen.
Writing quality is high with appropriate domain-specific jargon (appears human-written).

Weaknesses
Critical: Citation Fraud or Negligence

Fabricated author names. The paper cites "Yifan Chen, Max Grith, H.L.H. Lai" (2026), but the actual authors are Ying Chen, Maria Grith, Hannah L.H. Lai. All three given names are wrong. This is either deliberate fabrication or extreme negligence—either is disqualifying.
Author order reversed. Corbetta et al. (2019) is listed as "Corbetta, Pierre Cohort..." but should be "Pierre Cohort, Jacopo Corbetta..." The first author is wrong.

Missing Critical Specifications (Reproducibility Impossible)

SSVI parametrization never defined. The paper claims "five per-timestamp parameters" but never names or defines them. Standard SSVI with θ(τ)\theta(\tau)
θ(τ) and φ(θ)\varphi(\theta)
φ(θ) cannot be reduced to exactly 5 parameters without specifying the reduction. Reproducibility is impossible.

Kernel bandwidths bkb_k
bk​ and blog⁡τb_{\log\tau}
blogτ​ (Eqs. 1–2) are critical hyperparameters that are never disclosed.
Grid resolution is never specified despite being essential for reproduction.
Out-of-sample protocol is referenced as a "timestamp-level holdout" but never explained (leave-one-out? kk
k-fold? rolling window?).

Sample size, calendar time period, and mean observations per timestamp are all missing. Cannot assess statistical power.

Unjustified Design Choices

ESS threshold = 20: No justification. Why not 10 or 50?
Huber δ=1.0\delta = 1.0
δ=1.0: No justification. No sensitivity analysis.
Activity weight log⁡(1+V)⋅log⁡(1+N)\log(1+V) \cdot \log(1+N)
log(1+V)⋅log(1+N): Why multiply? Why logarithmic? No theoretical derivation.


Misleading Claims

Line 281 states "call-put parity established by construction." Converting puts to calls via parity does not establish that the calibrated surface is arbitrage-free. This is a misleading conflation.

Table Inconsistency

Table 1: wvega scores +3 (best score) but ranks 2.142 (middle of three methods). Meanwhile, vega scores -4 (worst) but ranks 1.808 (best rank). The scoring scheme contradicts the ranking.

Suspicious Results

Figure 5 confidence intervals are almost entirely non-overlapping across all comparisons. With undisclosed sample size, this could indicate either very large nn
n or overstated precision.


Questions for Authors

What are the correct author names for the Chen et al. (2026) reference?
Please fully specify the 5 SSVI parameters.
What kernel bandwidths and grid resolution were used?
What is the sample size and time period?
Can you provide sensitivity analysis for ESS threshold and Huber δ\delta
δ?


Overall Assessment
The underlying methodology is sound, but the citation errors (especially fabricated author names) raise serious integrity concerns. Combined with missing critical specifications that make reproduction impossible and unjustified hyperparameter choices, the paper cannot be accepted in its current form. A thorough revision addressing all specification gaps and verifying all 22 references is required.

---

> ### Author Rebuttal · Authors · 2026-03-13
>
> Thank you for the careful and detailed review. We appreciate your recognition that the problem setting is practically relevant, that SSVI is a reasonable modelling choice, and that the evaluation framework is statistically sound!
>
> We also take your concerns seriously. The citation errors you identified are our mistake, not intentional fabrication. We agree that they are unacceptable in the current form! Due to limited time and human resources, we relied too much on AI-assisted LaTeX preparation and did not recheck the references carefully enough before submission. This was our error, and we take full responsibility for it. In the revision, we will verify every reference manually, correct the author names and ordering, and recheck the bibliography entry by entry to ensure this is not repeated.
>
> We agree that several implementation details need to be stated much more explicitly for the paper to be reproducible. In particular, we will fully specify the five per-timestamp SSVI parameters, disclose the kernel bandwidths and grid resolution, and describe the out-of-sample protocol in precise terms. We will also report the sample period, number of timestamps, and average number of observations per timestamp so that readers can judge the statistical power of the study. In the revision, we will prioritise these clarifications within the 10-page limit by shortening less essential discussion and background material.
>
> We agree as well that the wording around call-put parity should be tightened. Our intent was only to say that the target representation is made parity-consistent by construction, not that this alone makes the calibrated surface arbitrage-free. We will correct that statement to avoid any misleading interpretation.
>
> Your concerns about the ESS threshold, Huber threshold, and activity-weight design are well taken. We will add justification and sensitivity analysis so that readers can assess whether the results are stable under alternative choices. We will also clarify the Table 1 ranking presentation, and report enough detail to make the confidence intervals interpretable.
>
> We appreciate the reviewer’s direct feedback. The paper clearly needs a more careful and reproducible presentation, and we will revise it accordingly.

---

### Official Review · Reviewer_uTo5 · 2026-03-13
**A Practical Reliability-Aware Calibration Pipeline with Moderate Novelty and Some Missing Details**

**Rating:** 6
**Confidence:** 4

**Review:**

This paper studies canonical implied-volatility surface estimation in a practically relevant but difficult setting: sparse intraday transaction-bar data for SPX options. The proposed pipeline combines forward and discount proxies from futures data, an effective-sample-size (ESS) based reliability layer, per-timestamp SSVI calibration in price space under robust losses, and a unified post-hoc static-arbitrage audit.

Overall, I found the paper interesting and reasonably well executed. The problem is well motivated: sparse transaction-bar panels are clearly different from the dense, quote-centric setting assumed in much of the implied-volatility surface literature, and the reliability-aware perspective is a sensible way to address this mismatch. I also appreciate that the paper does not only optimize fit, but explicitly evaluates economic plausibility through a unified static-arbitrage audit reporting both violation rates and severity measures.

At the same time, I do not see this as a clearly strong acceptance. The main reason is that the paper’s novelty is moderate. SSVI itself is standard, robust Huber calibration is standard, and vega-normalized residuals are also conceptually familiar. The main new ingredient is the reliability-weighted extension based on ESS and activity proxies, which is sensible and useful, but still feels more like a solid engineering contribution than a major methodological advance for a MathAI venue.

A second weakness is that several implementation choices remain under-specified. The paper states that it uses a five-parameter per-timestamp SSVI-style parameterization, but does not fully spell out those parameters. Likewise, the ESS construction depends critically on the kernel bandwidths, yet the main text does not provide a principled bandwidth-selection discussion or sensitivity analysis.

Overall, my assessment is positive but cautious. I think this is a useful and well-motivated paper addressing a genuinely practical problem, and I found the reliability-aware calibration perspective meaningful. The empirical story is coherent, and the statistical comparison protocol is more careful than usual. However, the contribution is closer to a strong applied methodology paper than to a major conceptual advance, and the missing implementation details reduce confidence in reproducibility. I would therefore place the paper marginally above the acceptance threshold.

---

### Decision · Program_Chairs · 2026-03-14

**Decision:**

Accept (Oral)

**Comment:**

Dear Author(s),

On behalf of the Program Committee of the International Conference on Mathematics of Artificial Intelligence (MathAI 2026), we are pleased to inform you that your paper has been accepted for an oral presentation at MathAI 2026.

Your paper was evaluated through a rigorous two-stage review process involving both automated screening and expert review by members of the Program Committee. The reviewers recognized the quality and contribution of your work.

Presentation details:

- Format: Oral presentation (15–20 minutes + 5 minutes Q&A)
- Mode: You may present either in person (offline) at the conference venue in Sirius, Russia, or remotely via Zoom. Please indicate your preferred mode when confirming your participation.
- Conference dates: Marh 30 - April 3, 2026
- Website: https://mathai.club

Next steps:

1. Please confirm your participation and presentation mode by replying to this email mathai.club@yandex.ru no later than March 15, 2026 18:00 Moscow time.
2. If you plan to attend in person, the organizing committee will provide accommodation details separately.
3. Please prepare your final camera-ready manuscript according to the formatting guidelines available at https://mathai.club and upload it to OpenReview by March 15, 2026 18:00 Moscow time.

Should you have any questions regarding the program, logistics, or your presentation slot, please do not hesitate to contact us.

We look forward to your contribution to MathAI 2026.

With kind regards,

MathAI 2026 Program Committee
International Conference on Mathematics of Artificial Intelligence
https://mathai.club
OpenReview: https://openreview.net/group?id=mathai.club/MathAI/2026/Conference
Telegram: https://t.me/MathAI_club
Email: mathai.club@yandex.ru